# Signature of Circulating Biomarkers in Recurrent Non-Infectious Anterior Uveitis. Immunomodulatory Effects of DHA-Triglyceride. A Pilot Study

**DOI:** 10.3390/diagnostics11040724

**Published:** 2021-04-19

**Authors:** Maria D. Pinazo-Durán, Jose J. García-Medina, Silvia M. Sanz-González, Jose E. O’Connor, Ricardo P. Casaroli-Marano, Mar Valero-Velló, Maribel López-Gálvez, Cristina Peris-Martínez, Vicente Zanón-Moreno, Manuel Diaz-Llopis

**Affiliations:** 1Ophthalmic Research Unit “Santiago Grisolía”, Fundación Investigación Sanitaria y Biomédica (FISABIO), Ave. Gaspar Aguilar 90, 46017 Valencia, Spain; dolores.pinazo@uv.es (M.D.P.-D.); jj.garciamedina@um.es (J.J.G.-M.); vavema@alumni.uv.es (M.V.-V.); 2Research Group in Cellular and Molecular Ophthalmo-Biology, Department of Surgery, University of Valencia, Ave. Blasco Ibañez 15, 46010 Valencia, Spain; manuel.diaz@uv.es; 3Spanish Net of Ophthalmic Research “OFTARED” RD16/0008/0022, of the Institute of Health Carlos III, 28029 Madrid, Spain; rcasaroli@ub.edu (R.P.C.-M.); maribel@ioba.med.uva.es (M.L.-G.); peris_crimar@gva.es (C.P.-M.); vczanon@universidadviu.com (V.Z.-M.); 4Department of Ophthalmology, General University Hospital “Morales Meseguer”, Ave. Marqués de los Vélez, s/n, 30008 Murcia, Spain; 5Department of Ophthalmology and Optometry, University of Murcia, Edificio LAIB Planta 5ª, Carretera Buenavista s/n, 30120 El Palmar Murcia, Spain; 6Laboratory of Cytomics, Joint Research Unit Principe Felipe Research Center and University of Valencia, 46010 Valencia, Spain; jose.e.oconnor@uv.es; 7Department of Surgery, School of Medicine and Hospital Clinic de Barcelona, University of Barcelona, 08036 Barcelona, Spain; 8Department of Ophthalmology, University Clinic Hospital of Valladolid, 47003 Valladolid, Spain; 9Ophthalmic Medical Center (FOM), Foundation for the Promotion of Health and Biomedical Research of Valencia (FISABIO), 46015 Valencia, Spain; 10Faculty of Health Sciences, Valencian International University, 46002 Valencia, Spain

**Keywords:** recurrent anterior non-infectious uveitis, cytokines, omega-3 fatty acids, triglyceride of docosahexaenoic acid

## Abstract

The purpose of this study was to identify circulating biomarkers of recurrent non-infectious anterior uveitis (NIAU), and to address the anti-inflammatory effects of triglyceride containing docosahexaenoic acid (DHA-TG). A prospective multicenter study was conducted in 72 participants distributed into: patients diagnosed with recurrent NIAU in the quiescence stage (uveitis group (UG); *n* = 36) and healthy controls (control group (CG); *n* = 36). Each group was randomly assigned to the oral supplementation of one pill/day (+) containing DHA-TG (*n* = 18) or no-pill condition (−) (*n* = 17) for three consecutive months. Data from demographics, risk factors, comorbidities, eye complications and therapy were recorded. Blood was collected and processed to determine pro-inflammatory biomarkers by bead-base multiplex assay. Statistical processing with multivariate statistical analysis was performed. The mean age was 50, 12 (10, 31) years. The distribution by gender was 45% males and 55% females. The mean number of uveitis episodes was 5 (2). Higher plasma expression of interleukin (IL)-6 was detected in the UG versus the CG (*p* = 5 × 10^−5^). Likewise, significantly higher plasma levels were seen for IL-1β, IL-2, INFγ (*p* = 10^−4^), and TNFα (*p* = 2 × 10^−4^) in the UG versus the CG. Significantly lower values of the above molecules were found in the +DHA-TG than in the -DHA-TG subgroups, after 3 months of follow-up, TNFα (*p* = 10^−7^) and IL-6 (*p* = 3 × 10^−6^) being those that most significantly changed. Signatures of circulating inflammatory mediators were obtained in the quiescent stage of recurrent NIAU patients. This 3-month follow-up strongly reinforces that a regular oral administration of DHA-TG reduces the inflammatory load and may potentially supply a prophylaxis-adjunctive mediator for patients at risk of uveitis vision loss.

## 1. Introduction

Uveitis is one of the major causes of vision loss in working-age people from developed countries [1]. The Standardization of Uveitis Nomenclature (SUN) working group [2] classified uveitis according to its localization in the eye globe as: anterior, intermediate, and posterior uveitis. Anterior uveitis is the most frequent intraocular inflammation in clinical practice and involves inflammation of the iris, ciliary body, or both. In addition, the infectious etiology should be considered in uveitis. Non-infectious uveitis (NIU) can be idiopathic, but can also result from immune-mediated processes or systemic disorders [2,3]. We also have to consider the laterality of uveitis because bilateral cases tend to be linked to chronic systemic processes and unilateral cases are likely acute, idiopathic, or infectious [1,2,3]. In general, autoimmune diseases follow a relapsing–remitting or a chronic progressive course. Chronic (or persistent) uveitis is known as the active process that endures longer than three months, with less severe signs and symptoms of uveitis, accompanied by manifestations of chronicity [1,2,3,4]. Acute (or limited) uveitis refers to cases with an episode lasting less than 3 months, displaying a wide variety of manifestations from moderate to very severe forms [1,2,4,5].

SUN defines recurrent uveitis as the condition in which the episodes are separated by at least three months of inactivity without treatment. On the contrary, any uveitis under steroid or immunosuppressive/immunomodulatory therapy remains active, and its socio-economic burden is high [3,6,7,8]. In this context, molecular information about recurrent non-infectious acute uveitis (NIAU) in the silent stage of the disease has not yet been reliably determined.

Lymphocytes are pivotal cells of the adaptive immune response. Two main lymphocyte types are the B cells (that secrete antibodies) and the T cells that migrate to the thymus and undergo a maturation process to destroy (T-killer cells) compromised cells, as well as to help alert (T-helper cells) other leukocytes. The T cells, also known as T-helper (Th) cells or CD4+ cells, coordinate the immune response, being the most important cytokine producers. These cells, in turn, can be subclassified into Th1, Th2 and Th17 cells that release different cytokine combinations (Figure 1).

The immune response in NIAU is orchestrated by a T cell-mediated autoimmune event, and chronically expanded by means of pro-inflammatory mediators, among them cytokines and chemokines [9]. A wide spectrum of pro-inflammatory cytokines and chemokines have been identified in humans and experimental NIAU models, and this knowledge has been reactivated over recent years [10,11,12], also in relation to ocular diseases [13,14,15]. 

Nutrition and nutritional control have been suggested as potential interventions for slowing the progression of several diseases, such as ocular surface dysfunction dry eyes (OSD), glaucoma, diabetic retinopathy (DR), or age-related macular degeneration (AMD), much of these with controversial results [16,17,18,19,20,21]. It has been shown that docosahexaenoic acid (DHA), an omega-3 polyunsaturated fatty acid (ω3 PUFA), and its anti-inflammatory derivatives (series-3 prostaglandins, resolvin D1, neuroprotectin D1, maresin, etc.) exert anti-inflammatory actions in a wide variety of cells and tissues [22,23,24,25,26,27]. Epidemiological and experimental studies have demonstrated that the sustained intake of fish containing high amounts of ω3 PUFA benefits the cardiovascular system [25] and the neurological–neurosensory organs [13,14,15,19,28,29]. 

The concerns of nutritional supplements in ophthalmology have been extensively discussed [13,14,15,16,17,18,19,20,21,28,29]. While nutraceuticals cannot replace eating in terms of food quality and quantity, they can benefit individuals suffering from poor nutrition, eating disorders, bariatric surgery, and/or malabsorption syndrome, among others [30,31,32]. In this regard, up to a half of Americans and many Europeans and Asians take at least one dietary supplement per day [20,21]. 

Fish oil supplements may contain ω3 PUFAS in triglyceride (TG) or ethyl ester forms. The TGs are the molecular forms of all fats and oils in nature [33]. TG is made of three fatty acids that are esterified to a glycerol backbone, providing protection to the double bonds from being exposed to oxidative injury. When single fatty acids are esterified to an ethanol backbone, this ethyl ester form is less stable and more vulnerable to oxidative attack. It has been shown that TG containing DHA (DHA-TG) attenuated nitric oxide production and suppressed the activation of nuclear factor-κβ, which induces the expression of pro-inflammatory cytokines (interleukin (IL)-6 and tumor necrosis factor (TNF-α) in lipopolysaccharide (LPS)-stimulated BV-2 microglial cells [30]. A daily DHA-TG supplementation (lasting from 3 months to 3 years) improved total antioxidant capacity and reduced serum IL-6 in patients suffering from pseudoexfoliation glaucoma [34], non-proliferative DR [19,35] or diabetic macular edema [36] when compared to non-supplemented controls. DHA-TG supplementation has also been shown to reduce the plasmatic levels of inflammatory markers IL-6 and IL-1β after exercise in endurance athletes [37], as well as in tears from patients suffering dry eye [13,14,15]. 

Further research is needed to understand the clinical and molecular bases for relapse following an episode of NIAU so as to better manage the disease. Therefore, the main scope of the present study was to establish the signature of circulating biomarkers of inflammation in a cohort of patients with inactive recurrent NIAU. The secondary objective was to assess the effects of the oral administration of DHA-TG on the clinical and molecular biomarkers of the above uveitis cohort, after a three-month follow-up.

## 2. Materials and Methods

### 2.1. Study Design 

The present study was done in accordance with the recommendations of the Code of Ethics of the World Medical Association (Declaration of Helsinki; Edinburgh, 2000), the EC normative, the ARVO guidelines and the Ethics Committee standards of the study centers (CEIC HUDP report no. 71/18), as well with the general and specific requirements for clinical research in order to maintain the privacy of the data obtained. 

A prospective interventional open-label multicenter study of patients in the quiescent stage of recurrent NIAU was carried out between 2018 and 2020 in Valencia, Barcelona, Murcia, and Valladolid (Spain), with the main purpose of outlining clinical characteristics, classical and specific blood-based parameters, and typical clinical hallmarks, in order to identify new molecular biomarkers for the disease. The second objective of this study was to assess the effect of a course of DHA-TG on the ocular state and on circulating biomarkers of the above patients.

The sample size was calculated; accepting an alpha risk of 0.05 and a beta risk of 0.2 in a two-sided test, 66 subjects were necessary to recognize as statistically significant a difference consisting of an initial proportion of 0.65 and a final proportion of 0.35. We anticipated a drop-out rate of 12%. We used the software Ene 3.0 (Servei d’Estadística Aplicada, Universitat Autonoma de Barcelona, Barcelona, Spain) for this calculation. Taking this calculation into account, 70 subjects were finally included in the study, who were classified into 2 groups: (1) the uveitis group (UG, *n* = 35), and (2) the control group (CG, *n* = 35).

### 2.2. Screening Procedures 

#### 2.2.1. Selection of Participants 

Patients who presented to the participating hospitals having previous episodes of NIAU, and who were determined to be in remission within at least 3 months of the last uveitis episode, were recruited for the study. A preliminary interview was done, according to the inclusion/exclusion criteria listed in Table 1. 

Remission was defined as inactive uveitis for at least 3 months without using corticosteroids/immunosuppressive/immunomodulatory treatments. Patients who did not reach remission, who were at high risk of relapse during the period of quiescence or suffered inflammation during the study follow-up were not eligible, or were excluded from the study. 

At baseline, 84 individuals were initially recruited, but only 70 suitable participants were included in this study, as reflected in Figure 2. Data were recorded as: socio-demographics, personal facts including duration of the disease since the first uveitis episode, intervals between episodes, familial uveitis background, current treatments, lifestyle characteristics and habits (drinking, smoking and nutrition facts), as well as systemic inflammatory diseases (such as those previously diagnosed and registered in history and examination findings). Active uveitis at the time of sampling and during follow-up was defined as positive Tyndall (trace cells in the aqueous humor) on BMC examination and one or more uveitis signs.

#### 2.2.2. Ophthalmic Examination

At the time of the first appointment for this study course, patients underwent a systematized ophthalmological examination in both eyes, including: logmar best corrected visual acuity (BCVA); biomicroscopy (BMC) and photographs of the anterior eye segment and media (ImageNet; Topcon, Barcelona, Spain); Goldman applanation tonometry (AT-900, Hagg-Streit, Harlow, Essex, UK) to assess the intraocular pressure (IOP); slit-lamp ocular fundus examination with 78 diopter lens and photographs of the posterior eye segment (ImageNet; Topcon, Barcelona, Spain). 

In accordance with previous history, all patients in the uveitis group (UG) were further evaluated to confirm the inactive recurrent NIAU diagnosis, according to the SUN working group criteria [2], as explained in Section 2.2.1. Selection of participants. Worsening was defined as an augmentation in the specific uveitis signs and symptoms during the study course.

#### 2.2.3. DHA-TG Supplementation Study 

From the initially recruited and examined individuals, 70 suitable participants of both sexes, aging 34–70 years, were distributed into the UG (*n* = 35) and the CG (*n* = 35). Each group was randomly assigned to oral supplementation with one pill at lunch (+DHA-TG) of DHA-TG per day (*n* = 18), or no supplementation (-DHA-TG) (*n* = 17) over three consecutive months (Figure 2). Pills containing DHA-TG, commercialized as Brudyitis^®®^ in packs of 30 capsules (Brudylab, Barcelona, Spain) (Figure 3), were kindly donated to the researchers by the manufacturer and given to the participants without any cost. A final ophthalmic visit was scheduled at the end of the 3-month follow-up with the same purpose and proceedings as the first appointment.

#### 2.2.4. Sampling Procedures

Blood sampling was scheduled at baseline and after 3-month follow-up in all participants. In fasting conditions (08.00 h a.m.), blood was collected from the antecubital vein into vacutainer tubes (Becton Dickinson, Auckland, New Zealand). One EDTA tube was used for biochemistry determinations at the clinical analysis department of the study centers. One sodium citrate tube was immediately centrifuged at 1500× *g* for 10 min using a refrigerated centrifuge at 4 °C to separate the plasma fraction, which was immediately aliquoted and stored at −80 °C until processing to specific assays at the laboratories of the Ophthalmic Research Unit “Santiago Grisolía”/FISABIO and the biochemistry laboratories of the Surgery Department at the Faculty of Medicine of the University of Valencia (Valencia, Spain). All experiments were done in duplicate. 

#### 2.2.5. Multiplex Determination of Plasmatic Cytokines

Flow cytometry (FCM) is a powerful tool that involves basic research, clinical diagnosis and therapy, and allows the analysis of single cells and molecules in the context of immunology and immunopathology [38,39]. A recent adaptation of FCM is the multiplex immunobead-based technique, which allows us to quantify simultaneously the levels of several cytokines/chemokines in a single sample [40,41,42,43,44]. This tool has proven advantages over other immunotechniques, such as enzyme-linked immunosorbent (ELISA), because of its higher sensitivity and the smaller sample volume (as in the case of ocular fluids) required for the assays [13,14,15]. 

The concentrations of the cytokines IL-1β, IL-2, IL-6, IL-8, IL-10 and IL-12, granulocytes and macrophages colony stimulating factor (GM-CSF), inferferon gamma (IFNγ), tumor necrosis factor alpha (TNFα) and vascular endothelial growth factor (VEGF), were analyzed in plasma samples with a Luminex^®®^ R-100 dedicated FCM system (Luminex, Austin, TX, USA), using a customized human cytokine/chemokine high-sensitivity custom panel (Procartaplex 10 plex; ThermoFisher Scientific, Labclinics, Barcelona, Spain) according to the manufacturer and previous reports [40,41,42,43,44]. In this assay, a cocktail of cytokine-specific polystyrene beads coupled covalently to cytokine-specific capture antibodies was allowed to react for 1 h at room temperature in a 96-well multiwell plate containing 30 µL of plasma sample per well. In other wells, a cytokine standard curve was constructed with appropriate dilutions of a standard solution containing known concentrations of all the tested cytokines. Two samples of each participant were used. Wells were washed by filtration to eliminate all unbound proteins, and a biotinylated detection antibody specific to a different epitope of each cytokine was added and incubated for 30 min at room temperature. The reaction mixture was revealed by means of streptavidin–phycoerythrin, which binds to the biotinylated detection antibodies. Finally, the intrinsic bead color and the fluorescence of the fluorescent label were used to identify and quantify the cytokine molecules. The cytokine concentrations of test samples were automatically calculated by the Bio-Plex Manager software from the cytokine standard curve, with assay range and sensitivity for each individual cytokine as summarized in Table 2. All the cytokines in the samples were within the linear range of the assay, except for IL-8 and IL-12, which were non-detectable. Data were expressed as mean and standard deviation (SD) in pg/mL.

### 2.3. Statistical Analysis

The “Microsoft Excel” program and the “IBM Statistical Package for the Social Sciences” SPSS Version 26.0 program (IBM SPSS Statistics for Windows, V 26.0. Armonk, NY, USA) were utilized. Qualitative variables were described by absolute or relative frequencies. The normality of the quantitative variables was verified using the Shapiro–Wilk test. Subsequently, the normal quantitative variables were described using the mean (SD), while the non-normal quantitative variables were described using the median and interquartile range. Finally, a parametric (Student’s t for independent or related samples) or non-parametric (Mann–Whitney or Wilcoxon U) test was applied for the comparison of two means. A *p* value less than 0.05 was considered statistically significant. 

## 3. Results

The mean age of the study participants was 50 (10) years (47 (9) years for the CG and 53 (11) years for the UG). The distribution by gender was 45% men and 55% women. At baseline, 70 participants were included in the study protocol, with 62 finally completing the follow-up (87.5%). Among the latter, 32 patients pertained to the UG (composed of recurrent NIAU patients in the quiescent stage of disease), and 30 individuals belonged to the CG (see the flowchart in Figure 2). The causes of withdrawal (12.5%) were: (a) Patients suffering one uveitis episode during the study course (three patients). (b) Problems with the manipulation and processing of blood samples (two cases). (c) Necessity of treatments that could interfere with the study results (two participants). (d) One participant abandoned the study voluntarily.

From the initial 70 suitable participants, 35 patients were classified as having inactive recurrent NIAU, with duration of disease ranging from 22 to 80 months, and the interval between episodes ranging from 15 to 76 weeks. In total, 52% of the subsequent NIAU episodes occurred in the same eye as was previously affected. Patients undergoing a long-term uveitis course had a greater number of episodes (more than six) than patients with shorter uveitis duration (*p* = 0.002). Data from demographics and risk factors are summarized in Table 3.

The precise examination of the anterior eye segment involved the biomicroscopic observation of the cornea aqueous humor, iris, pupil and lens. As expected in the recurrent NIAU patients, which refers to relapsing inflammation separated by >3 months without treatment, no signs of active anterior eye segment inflammation were seen, confirming the quiescent stage of disease. Posterior synechiae (adhesions between the anterior lens surface and the anterior iris) were detected in 18% or the UG (Figure 4A). Lens pigment deposition and patchy clumps on the anterior lens surface (27%), as well as lens opacity (32%), were also seen. Few cases of more extensive synechiae in the rest of the fibrinous membranes of the pupil were detected (10%) (Figure 4B). Idiopathic isolated non-granulomatous anterior uveitis was previously diagnosed in 63% of our study cases, with up to a half of these having seasonal variation. Bilateral presentations (37% of our UG patients) tend to be associated with chronic systemic disorders, while unilateral uveitis was observed in the idiopathic uveitis patients of our study cohort. Ocular fundus examination and retinographies did not reveal any significant pathologic sign related to posterior uveitis in the study participants. Moreover, no relevant pathology of the retina and optic nerve was detected in any of the study groups (Figure 4C,D). 

The main purpose of this study was to investigate the underlying immunopathogenesis of the quiescent stage of recurrent NIAU, by assaying the signature of inflammatory circulating biomarkers in these patients. The cytokine/chemokine panel assayed in this study included the following: IL-1β, IL-2, IL-6, IL-8, IL-10, IL-12, GM-CSF, IFNγ, TNFα, and VEGF. Up to 90% of the molecules from the above assayed panel were identified in blood samples of the study participants. Two molecules were not detectable (IL-8 and IL-12).

At baseline, the UG patients showed significantly higher plasma levels of IL-6 than the CG, constituting the most significant differentially expressed molecule between our study participants (*p* = 0.00005). Additionally, IL-1β, IL-2 and IFNγ (*p* = 0.0001), as well as TNFα (*p* = 0.0002), IL-10 (*p* = 0.003) and GM-CSF (*p* = 0.005), displayed significantly higher levels in the UG than in the CG. VEGF plasma expression levels did not reveal statistically significant changes between groups. The above results are summarized in Figure 5.

The secondary objective of this study was to investigate the effects of the oral administration of DHA-TG during three consecutive months on the clinical and molecular biomarkers of our participants. In this context, the UG and CG patients were homogeneously divided into those randomly assigned to a daily pill (+DHA-TG group), and those assigned to no-pill (-DHA-TG group), of the DHA-TG commercialized nutraceutic (Figure 2 and Figure 3). 

The scarce uveitis episodes occurring during the study course (8.5%) have to be considered against the background of DHA-TG intake. Up to half of the +DHA-TG group of UG patients also reported subjective sensations regarding amelioration of joint pain and fatigue, improvement of mood, memory, and vision, as well as positive effects on the skin, ankles and hair. No adverse effects were reported during the study course resulting from the oral administration of the DHA-TG product. 

According to the comparison of the subgroups, DHA-TG supplementation caused significant changes in the signature of plasma pro-inflammatory mediators at the end of the 3-month follow-up (Figure 6). Data showed a significant reduction in the plasma levels of the main assayed molecules after 3-month follow-up. The results also pinpointed that TNFα (*p* = 10^−7^), IL-6 (*p* = 3 × 10^−6^), IL-10/INFγ/VEGF (*p* = 3 × 10^−5^), IL-1β (*p* = 8 × 10^−5^), IL2 (*p* = 10^−4^), and GM-CSF (*p* = 2 × 10^−4^) were the most significantly reduced pro-inflammatory mediators in the plasma of the +DHA-TG group versus the -DHA-TG group, thus demonstrating that the inflammatory load was noticeably lower in UG (see Figure 6) as a result of the supplement conferring a protective effect on recurrent NIAU patients in the quiescence stage of disease.

## 4. Discussion

The present study showed the different expression profiles of pro-inflammatory mediators in the blood samples of patients with recurrent NIAU (those with relapsing inflammation periods separated by more than 3 months without treatment) compared with the CG. Furthermore, our data showed significant changes in the signature of circulating cytokines/chemokines according to the DHA-TG regime. 

Uveitis is an important cause of eye and visual morbidity in people at working age, NIAU being the most common clinical type, which presents with a variable course [1,2,4,5,7,8,45]. The majority of NIAU cases are idiopathic, but it can be associated with a wide variety of ocular and systemic disorders [2,4,5,7,45,46,47]. 

Recurrent uveitis is defined as a process in which episodes are separated by at least three months of inactivity without treatment [1,5]. In our study, 35 patients were initially classified as having inactive recurrent NIAU, with a mean relapsing of 5 (2), as well as quiescent intervals between episodes ranging from 12 to 80 weeks. In the study of Grumwald et al. [48], the risk of relapse for cases of remitted primary anterior uveitis was much high than in ours. In this regard, patients should be advised of the risk of recurrence, so that they can acknowledge the symptoms of relapses and receive rapid and effective eye care. 

Data from the participants were recorded, such as socio-demographics, characteristics and lifestyle, including drinking and smoking habits and nutritional facts. The average age of our study cohort was 47 (9) years for the CG and 53 (11) years for the UG. The distribution by gender was quite similar between groups, with a higher percentage of women (45% of the controls and 53% of the UG). Our study cohort was only composed of a single ethnicity (Caucasians). A recent review performed from 1976 to 2017 on observational data from patients with uveitis in Europe [49] revealed that anterior uveitis accounted for 53.2% of 24,126 uveitis patients from studies performed in 20 European countries, including Spain, the mean age of the patients being 40 years (ranging 31 to 48 years) and women accounting for 52% of these patients, similarly to the data presented herein. In total, 990 cases of uveitis were identified in Italy from 2013 to 2015, these being mostly females (59%) with a median age at presentation of 44 years, and anterior uveitis was reported as the most frequent clinical type (53.5%) [50]. Brydak-Godowska et al. [49] analyzed data from 26 published studies with a total of 24,126 uveitis patients from 12 European countries, showing that idiopathic uveitis was the most common form (55.5%), while in our study idiopathic uveitis accounted for 60% of the cohort.

Other baseline characteristics of our uveitis patients were the higher percentages of monocular affectation (63%) compared to bilateral presentation (37%). Regarding familial uveitis background, only one case was recorded among our study participants. Other reports [51,52,53] claimed that in a large population-based uveitis study in the United States, the incidence was affected by aging, with women showing a higher prevalence of uveitis than men, as reported herein.

Th cells, the main regulators of adaptive function, serving as effectors for cell-mediated immunity, can also be subclassified into Th1, Th2 and T17 cells, which produce cytokines [9,54,55,56]. In this context, Th1 cells are the primary sources of the inflammatory cytokines IL-2, IL-10, lFNγ, and TNF-β, the most important being IFNγ. Th2 cells release IL-4, IL-5, IL-10, and IL-13, while Th17 cells produce IL-17, IL-22, IL-24, and IL-26 cytokines (see Figure 1). Cytokines are synthesized during the effector stage of innate and adaptative immunity to mediate the immune response [9,54,55,56]. An optimum balance between Th1 and Th2 cytokine production depends on the immune challenge. A variety of reports confirm that cytokine expression is augmented in the plasma, serum, cerebrospinal fluid, or aqueous humor of NIAU patients, or in patients at risk of NIU, with respect to healthy controls [9,10,11,56,57,58,59].

In trying to determine the underlying immunopathogenesis of relapsing–remitting NIAU, we investigated the signature of circulating biomarkers of inflammation and immune response between recurrences in our study participants via the multiplex immunobead-based FCM technique. From a molecular viewpoint, our findings highlighted the key events occurring in recurrent NIAU patients, despite the condition remaining inactive. A comparison of the main study groups displayed significantly higher plasma levels of IL-6 at baseline in patients with recurrent NIAU, which constituted the most significant differentially expressed molecule between our main study participants (*p* = 0.00005). Furthermore, IL-1β, IL-2, INFγ, TNFα, IL-10, and GM-CSF also exhibited significantly higher levels in the UG than in the CG. Takase et al. [58] pinpointed the cytokine profile in the aqueous humor and sera of infectious versus non-infectious uveitis patients, with IFN-γ being the most abundant cytokine in both uveitis types, followed by IL-10, similarly to our results (see Figure 5). However, in that report, other serum cytokines were under the detectable levels in both uveitis types, whereas in our study only two cytokines (IL-8 and IL-12) were undetectable. Interestingly, these authors emphasized that infectious uveitis is characterized by cytokines produced within the eyes, whereas the NIU is characterized by Th1 cytokines being produced by both the eyes and the circulating T cells. The authors concluded that human NIU is characterized by local and systemic Th1 cytokine production, but not Th2 cytokine production. However, in our recurrent NIAU cohort, we also detected significantly increased plasma levels of IL-10 (an important inhibitor of the production of proinflammatory cytokines, such as IL-1β and IL-6). The cytokines that we detected in the plasma of the study participants was significantly increased in the recurrent NIAU patients in its inactive stage, and several of these cytokines mainly appear in relation to natural killer (NK) cells, such as IL-6, which plays an important role in NK cells activation, or IL-10, which can be released from activated NK cells, as previously reported [60,61]. Furthermore, it has been shown that CD4 and CD8 T cell surface molecules are essential for the recognition and activation of T cells by binding to their corresponding class II/class I major histocompatibility complex ligands on an antigen-presenting cell. In fact, CD4 and CD8 can increase the antigen-induced IL-2 formation through distinct mechanisms [60,61,62,63]. Additionally, CD4+ T cells can promote NK cells and macrophages via INFγ, to kill host cells in certain conditions. In this regard, Sanz-Marco et al. [63] analyzed via FCM the CD4/CD8 ratio in the aqueous humor of patients undergoing an acute episode of uveitis, suggesting that the characterization of CD4 and CD8 in uveitis with T-lymphocyte involvement and difficult diagnosis (virus uveitis, sarcoidosis, masquerade syndromes in T lymphomas) may help in managing the affected patients. In our study cohort we identified the plasma signature of cytokines, IL-6 being the most significantly expressed. IL-6 is a soluble mediator with a variety of effects mainly regarding inflammation and immune response [9]. This cytokine is synthesized locally from the macrophages, during the initial phase of inflammation, but also by the endothelial cells. IL-6 has been shown to stimulate the production of fibrinogen and the C-reactive protein, leading to changes in erythrocyte and platelet formation, as well as stimulating the specific differentiation of naïve CD4+ T cells [64]. The IL-6 receptor (IL-6R)–signaling system is composed of two receptor chains and its downstream signaling molecules [9,64,65]. Epidemiological and experimental studies have demonstrated the role of IL-6 in disease [66,67]. Moreover, IL-6 blockade by gene therapy or therapeutic suppression (anti IL-6/anti IL-6R) has also been reported [68]. Significant differences in IL-6 plasma expression in the UG versus the CG (*p* = 5 × 10^−5^) were found in our study. We propose IL-6 as a biomarker of the inflammatory load of patients in the interplay between NIAU recurrences, with excellent potential for diagnosis, prognosis and therapy, for the better management of the NIAU patients. The management of NIAU is challenging. Even so, much remains to be done to definitively establish the relationships between IL-6 and other cytokines and chemokines in recurrent NIAU. Furthermore, pharmacological substances such as the anti-IL-6 monoclonal antibody need further exploration in order to reduce disease activity and to lengthen the time between episodes

Nutrition interventions for chronic disease prevention and prognosis have been extensively reported [25,26,27,28,29,30,31,69,70,71]. However, there is a lack of consensus among ophthalmologists and researchers on the interest of nutritional supplements containing antioxidant vitamins, carotenoids, trace elements and/or ω3 PUFAs in ophthalmology. Dry eye disease [14,15,16], myopia complications and progression [72], cataracts [32,73], or sight-threatening diseases such as glaucoma, DR, and AMD [16,17,18,19,20,21,32,74,75,76,77,78,79], have been important targets of antioxidants and ω3 PUFAs. DHA is an essential structural constituent of cell membranes. DHA is also a precursor for the biosynthesis of a wide variety of substances, among them the series-3 prostaglandins, resolvin D1, neuroprotectin D1, maresin, and others, for the exertion of their anti-inflammatory effects [22,23,24,25]. Diverse experimental and epidemiological studies showed that eicosanoids derived from arachidonic acid were reduced when experimental animals [80,81] and patients of chronic inflammatory diseases [82] were supplemented with EPA and/or DHA. In fact, the duration and dose of the supplementation are pivotal to the efficacy of long-chain ω3 PUFAs in chronic diseases [82]. 

In this scenario, the secondary end-point of the present work was to explore the effects of the supplementation regime of one daily pill of DHA-TG during 3 consecutive months on the clinical and molecular features of our participants. Significant changes in the profile of plasma pro-inflammatory mediators at the end of 3-month follow-up were observed according to the DHA-TG regime. A significant decrease in plasma levels of the main assayed molecules (in order, from highest to lowest: TNFα, IL-6, IL-10, INFγ, VEGF, IL-1β, IL2, and GM-CSF) was detected in the UG versus the CG at the end of study. It has been previously reported that a daily DHA-TG supplementation significantly reduces (a) the tear levels of IL-1β, IL6, and IL-10 in dry eye disease patients [13,83]; (b) the tear concentrations of IL-6 and TNF-α in chronic glaucoma patients [14], and (c) the tear amounts of IL-1b and IL-6 in women employees working with computers [15]. Moreover, a daily DHA-TG core nutritional supplement regimen significantly reduced IL-6 in the serum of pseudoexfoliation glaucoma patients [31], non-proliferative DR [32] or diabetic macular edema [33], plasma IL-1β and IL-6 after exercise in endurance athletes [34], and the TNF-α in the plasma of middle- and long-distance running athletes [83]. 

It has been suggested that smoking increases the risk of relapse of inflammatory and autoimmune diseases, in general [84,85,86]. Lee et al. [87] reported that smoking and Behçet disease were closely associated in a Korean population-based study. Costa et al. [88] also reported that smoking was associated with the risk of uveitis in Portuguese patients with spondyloarthropathies. Zhao et al. [89] extensively reviewed the role of tobacco habits in spondylarthritis. Moreover, Zanón-Moreno et al. [90] reported in chronic glaucoma that the IL-6, caspase-3, and PARP-1 levels were significantly higher in the plasma and aqueous humor of smoking elder women than in ex-smoking and non-smoking women of the same age. However, all studies remain inconclusive. Smoking habit is a potential risk factor for eye diseases, and inflammation mediators increase with smoking in the aqueous humor and plasma samples. Since smoking may be a confounding factor for plasma cytokine expression, we initially excluded all current smokers, ex-smokers and passive smokers.

It has been said that current knowledge on cytokines in uveitis is largely based on animal models. Due to this, we firmly believe that human studies are needed to improve practical knowledge for better uveitis eye care.

In summary, the underlying mechanisms of relapsing in NIAU are unclear. However, we could speculate that the key events occurring in the quiescent phase involve an outstanding increase in circulating inflammatory mediators. We may summarize that the upregulation of inflammatory cytokines plays an important role in the pathophysiology of uveitis, mainly IL-6, as well as IL-1β, IL-2, INFγ, TNFα, IL-10, and GM-CSF. We also demonstrated that regular DHA-TG supplementation noticeably reduces the inflammatory load, and may be useful for dietary intervention in recurrent NIAU patients at risk of blindness.

The limitations of the present study derive mostly from the partial retrospective nature of this study. In fact, incomplete follow-up or insufficient registers may have led to a mis-estimation of data. The relatively small final sample of participants after being distributed into each subgroup after the 3-month follow-up also has to be considered during the statistical processing. Patients who developed exacerbation of the NIAU were excluded from data processing in order to avoid statistical power loss (three cases). The above patients needed to be urgently treated. Relapse was not identified every time in the follow-up visits. Systemic conditions could be underestimated despite the exhaustive anamnesis, interspecialist visits and complementary exams. Regarding the blood collection and processing, small problems emerged (two cases). As cytokine levels have been shown to be noticeably different between serum and plasma, but cytokine levels in plasma were reported to be more stable than in serum, the study was run in plasma from UG patients versus CG. Furthermore, as the cytokines in serum and plasma are affected by multiple freeze/thaw cycles, special attention was paid to this concern by performing the cytokine assays at the same time to establish a simultaneous comparison between groups and subgroups [91]. Overall, further research requires a larger sample size and maximized efficiency in the protocols, in order to ensure high-quality results to validate the current data.

## 5. Conclusions

Our study demonstrated the different expression profiles of cytokines/chemokines in tears from recurrent anterior NIAU patients compared with healthy controls. We confirm that inflammation is present in the quiescent stages of uveitis, IL-6 being the most outstanding cytokine in this process. We propose that IL-6 represents an important challenge as a presumed prognostic biomarker for recurrent anterior NIAU. Furthermore, this 3-month follow-up work strongly reinforces that a regular oral administration of DHA-TG reduces the inflammatory load and may potentially supply an immunomodulatory prophylaxis adjunctive mediator for patients at risk of uveitis vision loss.

## Figures and Tables

**Figure 1 diagnostics-11-00724-f001:**
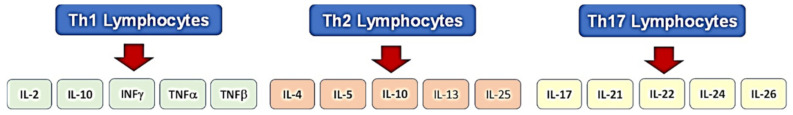
Th cells (Th1, Th2 and Th17) are the primary sources for the inflammatory cytokines.

**Figure 2 diagnostics-11-00724-f002:**
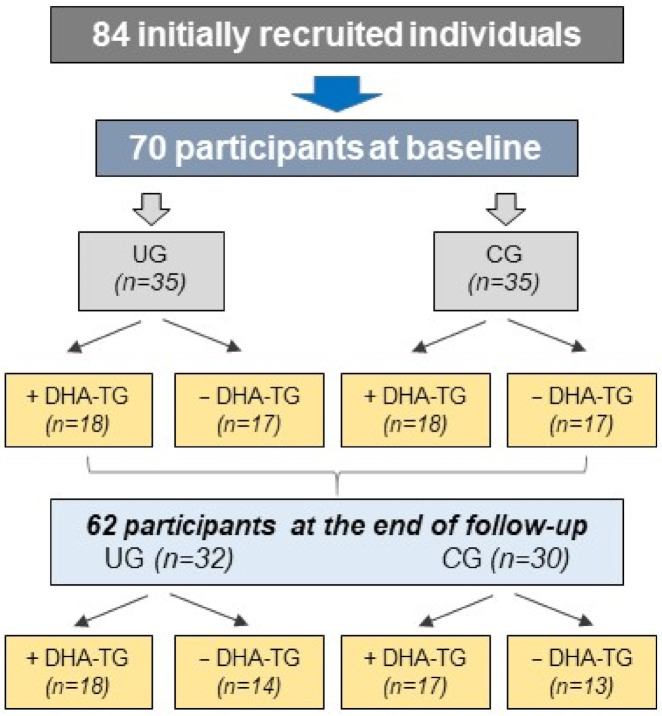
Flowchart of the study selection process and participants, at baseline and end of study. UG: uveitis group; CG: control group; +/− DHA-TG: (with/without) triglyceride containing w-3 docosahexaenoic acid.

**Figure 3 diagnostics-11-00724-f003:**
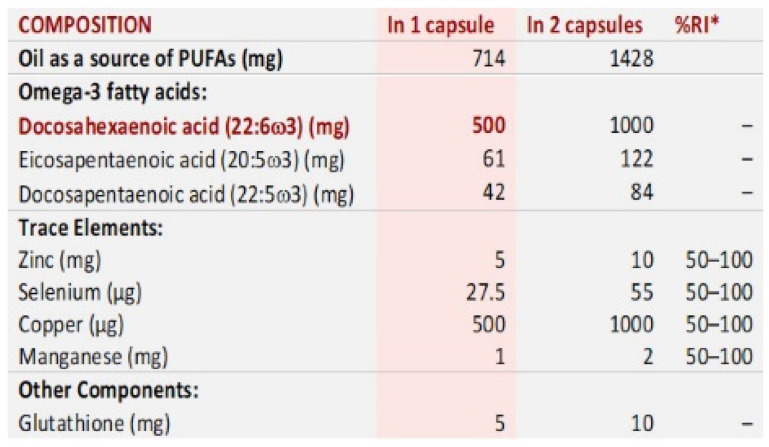
Composition of the nutritional supplement Brudyitis^®®^. * RI: reference intake for 1–2 capsules. ω-3: omega 3; PUFAs: polyunsaturated fatty acids; TG: triglyceride; DHA-TG: enzymatically re-esterified DHA.

**Figure 4 diagnostics-11-00724-f004:**
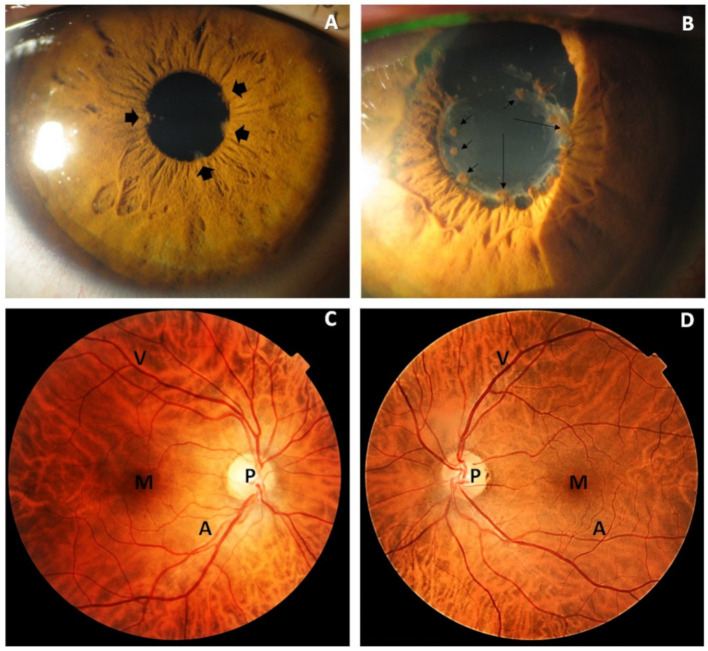
Biomicroscopic photographs (slit-lamp) of the anterior and posterior eye segment from patients with inactive recurrent non-infectious anterior uveitis (NIAU). (**A**) Posterior synechiae (arrowheads) resulted from focal adhesions between the posterior iris and the anterior lens surface. (**B**) Irregularly shaped pupil with posterior synechiae extended over the lens surface, with pupillary/prelental membranes (thin arrows) and lens deposits (small arrows). (**C**,**D**) Normal ocular fundus photographs of the right and left eyes of a 68-year-old patient with recurrent anterior NIAU uveitis in its quiescent stage. T P: optic disc papilla; M: macula; V: retinal vein; A: retinal artery. A deeply pigmented retina gives the appearance of a tessellated non-pathologic fundus (tigroid stripes).

**Figure 5 diagnostics-11-00724-f005:**
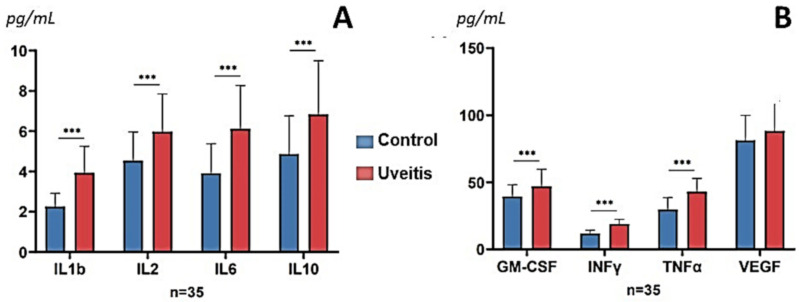
Plasma levels of pro-inflammatory mediators in the main groups of the study participants. Results show mean (standard deviation). (**A**) Interleukines. (**B**) Other cytokines. IL: interleukin; GM-CSF: granulocite macrophages colony stimulating factor; INF-γ: interferon gamma; TNF-a: tumor necrosis factor alpha; VEGF: vascular endothelial growth factor. Statistically significant *** = *p* < 0.0001).

**Figure 6 diagnostics-11-00724-f006:**
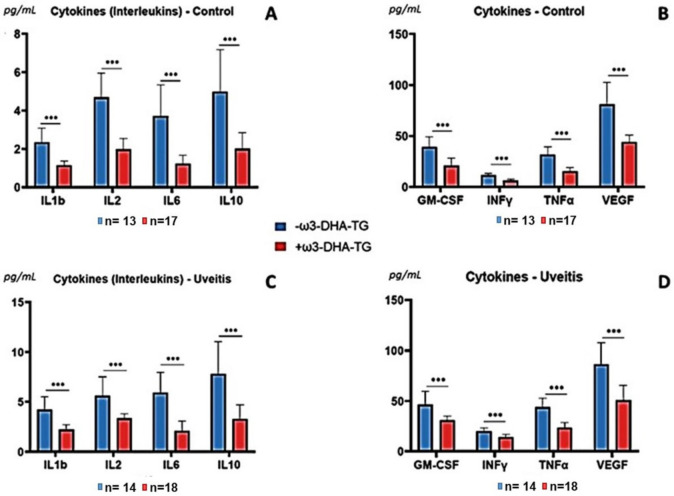
Differential expression profiles of plasma pro-inflammatory mediators in recurrent non-infectious anterior uveitis patients in its quiescent stage and healthy controls, according to the triglyceride form of docosahexaenoic acid regime. (**A**) Plasma level of interleukines in the control group, (**B**) plasma level of other cytokines in the control group, (**C**) plasma level of interleukines in the uveitis group, (**D**) plasma level of other cytokines in the uveitis group. Interleukins: IL -1β, -2, -6, -10; Granulocyte and macrophage colony stimulating factor: GM-CSF; interferon gamma: INFγ; tumor necrosis factor alpha: TNFα; vascular endothelial growth factor: VEGF; with/without omega-3 docosahexaenoic acid: +/− w3-DHA-TG. Levels are expressed as the mean in pg/mL, and bars represent the standard deviation. Statistically significant *** (*p* < 0.0001).

**Table 1 diagnostics-11-00724-t001:** Inclusion and exclusion criteria for the study participants.

Inclusion	Exclusion
Caucasian individuals aged older than 20.	Caucasian individuals over 70 years of age Non-Caucasian individuals.
Accurate diagnosis of recurrent NIAU (quiescent stage) for the corresponding uveitis group (UG).	Any other ocular disease than recurrent anterior non-inflammatory uveitis. Patients receiving local treatment that may interfere with the study (including topic nutraceutics). Eye laser/surgery in the previous 12 months.
Healthy individuals for the control group of participants (CG).	Patients experiencing any systemic disease, receiving local or systemic treatment that may interfere with the study (including oral nutraceutics). Surgery in the previous 12 months.
Individuals without disorders of substance use.	Smoking and/or drinking habits. Drug addiction.
Precise and complete data of the medical history. Those who could participate in the study and were able to do so.	History including diagnoses that do not fit with the study purpose. Unfeasibility of having a thorough and complete clinical history. Unable to participate.

NIAU: non-infectious anterior uveitis; UG: uveitis group; CG: control group.

**Table 2 diagnostics-11-00724-t002:** Sensitivity and range of the individual cytokine standard curves used in the multiplex assay.

Cytokine	Assay Sensitivity (pg/mL)	Assay Range (pg/mL)
**IL-1 β**	0.10	0.32–1330
**IL-2**	0.19	0.44–1820
**IL-6**	0.038	1.03–4230
**IL-8**	0.30	0.217–890
**IL-10**	0.038	0.18–750
**IL-12**	0.04	6.84–28,000
**GM-CSF**	1.30	1.79–7330
**IFNγ**	0.075	0.31–1250
**TNFα**	0.40	8.54–35,000
**VEGF**	0.88	5.86–24,000

Data are provided by the manufacturer and expressed as picograms/mL. Interleukin (IL); granulocyte macrophage colony-stimulating factor (GM-CSF); inferferon gamma (IFNγ); tumor necrosis factor alpha (TNFα) and vascular endothelial growth factor (VEGF).

**Table 3 diagnostics-11-00724-t003:** Participant characteristics.

Demographics and Characteristics	CG	UG
Age (years)	47 (9)	53 (11)
Gender (% male/female)	44/56	47/53
Laterality (% one/two)	-	63/37
Ethnicity	Caucasians	Caucasians
Idiopathic uveitis (%)	-	60
Duration of disease (months)	-	43 (20)
Number of episodes	-	5 (2)
Familial uveitis background (%)	-	2.8

Results are showed as mean (standard deviation). CG: control group; UG: uveitis group.

## Data Availability

Data from this study are contained within this article.

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
