# Peer review of "Signature of Circulating Biomarkers in Recurrent Non-Infectious Anterior Uveitis. Immunomodulatory Effects of DHA-Triglyceride. A Pilot Study"

_diagnostics, 2021, doi:10.3390/diagnostics11040724_

Round 1

Reviewer 1 Report

This manuscript contains clinically important data which support that administration of omega-3 docosahexaenoic acid has an anti-inflammatory effect in patients suffering from recurrent non-infectious anterior uveitis. Revisions were made by the Authors; therefore, I recommend the revised version of the manuscript for publication.

Author Response

This manuscript contains clinically important data which support that administration of omega-3 docosahexaenoic acid has an anti-inflammatory effect in patients suffering from recurrent non-infectious anterior uveitis. Revisions were made by the Authors; therefore, I recommend the revised version of the manuscript for publication.

We want to thank the Reviewer, for positive criticism to improve our ms, and for the high consideration to this work. Thank you so much for your help.

Reviewer 2 Report

The revised version of the Ms is an improvement and I have no hesitation to confirm my willingness of reading the final, published version of this very interesting pilot trial.

However, there are a few points that need the attention of the Authors in order to enhance the impact of their research work:

  • The Authors have structured a very long response to my question on sample size calculation and I would like to thank them for the effort. However, I did not questioned the inclusion or exclusion criteria that in no instance can be proposed as instrumental to sample size calculation. Indeed, in the revised Ms the Authors report “Sample size calculation (the appropriate number of individuals chosen for a study from the total 124 population) performed by means of the Ene 3.0 program, developed by the Servei d'Estadística Aplicada (Univer-125 sitat Automoma de Barcelona, Spain) and distributed by GlaxoSmithKline labs., according to the objectives of the study, 126 with a 95% confidence interval.” but this is not entirely correct nor it is sufficient to allow reproduction of the calculation. To this end the Authors are invited to incorporate the formula in the text with the whole set of parameters used. This is normally done in clinical trials in view of the request of the approving body (local, national or international regulatory board). In addition, the sentence “(the appropriate number of individuals chosen for a study from the total 124 population)” does not help clarify the issue. In fact, the sample size calculation is needed in relation of the outcomes set, no matter whether or not there this is the whole protocol analysis or a subset of analysis.
  • The second point is concerned with the measurement and report of the cytokines. I am very much interested in your data because of their blood origin. In this respect they are very important and deserve very much attention. In fact, the methodology used is highly sensitive and minute amount can be dected. Therefore, I would suggest to incorporate on top of Fig. 5 and Fig 6 the relative control curve to confirm that the valus are within the calibration range in all instance. In addition, both figures 5 and 6 would better represent the results by reporting dotted data from individual subject per single cytokine and experimental condition. Also, please notice that the actual legends do not reflect the heading of the histograms.
  • Finally, this is a pilot/preliminary interventional trial and this should be reflected in the title to make sure that the reader is informed about the need of a fully powered trial to conclude on the subject.  

Author Response

The revised version of the Ms is an improvement and I have no hesitation to confirm my willingness of reading the final, published version of this very interesting pilot trial.

However, there are a few points that need the attention of the Authors in order to enhance the impact of their research work: 

  • The Authors have structured a very long response to my question on sample size calculation and I would like to thank them for the effort. However, I did not questioned the inclusion or exclusion criteria that in no instance can be proposed as instrumental to sample size calculation. Indeed, in the revised Ms the Authors report “Sample size calculation (the appropriate number of individuals chosen for a study from the total 124 population) performed by means of the Ene 3.0 program, developed by the Servei d'Estadística Aplicada (Universitat Automoma de Barcelona, Spain) and distributed by GlaxoSmithKline labs., according to the objectives of the study, with a 95% confidence interval.” but this is not entirely correct nor it is sufficient to allow reproduction of the calculation. To this end the Authors are invited to incorporate the formula in the text with the whole set of parameters used. This is normally done in clinical trials in view of the request of the approving body (local, national or international regulatory board). In addition, the sentence “(the appropriate number of individuals chosen for a study from the total 124 population)” does not help clarify the issue. In fact, the sample size calculation is needed in relation of the outcomes set, no matter whether or not there this is the whole protocol analysis or a subset of analysis.

We thank the Reviewer for arising again concerns to this point. The explanation regarding the sample size calculation has been modified, according to the Reviewer suggestions, as follows:

“The sample size was calculated accepting an alpha risk of 0.05 and a beta risk of 0.2 in a two-sided test, 66 subjects were necessary to recognize as statistically significant a difference consisting in an initial proportion of 0.65, and a final proportion of 0.35. It has been anticipated a drop-out rate of 12%. We used the software Ene 3.0 (Servei d'Estadística Aplicada, Universitat Automoma de Barcelona, Spain) for this calculation Taking this calculation into account, 70 subjects were finally included in the study, who were classified into 2 groups: 1) uveitis group (UG, n = 35), and 2) control group (CG, n = 35).”

  • The second point is concerned with the measurement and report of the cytokines. I am very much interested in your data because of their blood origin. In this respect they are very important and deserve very much attention. In fact, the methodology used is highly sensitive and minute amount can be dected. Therefore, I would suggest to incorporate on top of Fig. 5 and Fig 6 the relative control curve to confirm that the valus are within the calibration range in all instance. In addition, both figures 5 and 6 would better represent the results by reporting dotted data from individual subject per single cytokine and experimental condition. Also, please notice that the actual legends do not reflect the heading of the histograms.

We want to thank The Reviewer for calling our attention to this point. The answers to the comments and suggestions indicated by the reviewer are enclosed below:

- Regarding the first comment, we want to emphasize that as well pointed out by the Reviewer, the methodology used is very sensitive and allows the detection of cytokine concentrations in the order of pg/mL. According to the reviewer suggestion on the concordance of our calculated values regarding the calibration curve, we have considered more illustrative to include a new table (Table 2) that summarizes the essential parameters of sensitivity and concentration range for each individual cytokine, as they provide an idea of the detection limits and linearity parameters of the assay. In spite of this modifications, If the Reviewer considers yet essential to display all the 10 single standard curves, we would rather incorporate them into the manuscript as Supplementary Data.

-In the case of the second comment, we thank again the Reviewer for pointing out that our statement and illustrations on the measurement and report of the cytokines found in our study could be understood in this way. However, we have clarified the explanation (in part) in the previous answers to the Reviewer comments, as well as in the current answers. With all due respect to the Reviewer, we will prefer to focus on the two groups of participants (uveitis and controls) rather than to display the results of each subject for each cytokine and experimental condition (at baseline and after supplementation), that may confound the readers with so many data. We kindly ask The Reviewer to maintain the data in the former version.

- We have corrected the figure legends, as suggested, for a better understanding of this part of our work as follows:

The Figure 5: We have eliminated CG: Control group of participants; UG: uveitis group of patients

The Figure 6: We have added an abbreviation and rewritten the final paragraph to: With/without omega-3 docosahexaenoic acid +/-w3-DHA-TG. Levels were expressed as the mean in pg/mL, and bars represent the standard deviation

In addition, we have deleted "see figure 5" in line 311, keeping "see figure 6".

Also, we have indicated in the text that our intention was only to refer to the Figure 6 when explained the protective background to the supplemented group of the recurrent NIAU patients

  • Finally, this is a pilot/preliminary interventional trial and this should be reflected in the title to make sure that the reader is informed about the need of a fully powered trial to conclude on the subject.  

We would like to thank the reviewer for pointing this out. As he suggested, we have changed the title to include this aspect, as indicated below:

“Signature of circulating biomarkers in recurrent non-infectious anterior uveitis. Immunomodulatory effects of DHA-triglyceride. A pilot study.”

In addition, a new sentence on this matter, has been added at the end of the discussion section  lines 434-435:

“Overall, further research requires a larger sample size and maximising efficiency in the protocols to ensure high quality results to validate the current data”.

This manuscript is a resubmission of an earlier submission. The following is a list of the peer review reports and author responses from that submission.

Round 1

Reviewer 1 Report

This manuscript contains clinically important data which support that administration of omega-3 docosahexaenoic acid has an anti-inflammatory effect in patients suffering from recurrent non-infectious anterior uveitis. Therefore, I recommend this manuscript for publication with minor modifications.

Comments:

The Authors should indicate what *** means in Figures 5 and 6, in the corresponding figure legends.

There are grammatical or spelling errors in lanes 46, 62, 78, 109, 291, 313, 325, 466.

Reviewer 2 Report

The Ms by Pinazo-Duran et al is well written and reports on a prospective interventional open-label multicenter study of patients in the quiescent stage of recurrent NIAU carried out between 2018-2020 in Valencia, Barcelona, Murcia and Valladolid (Spain), with the primary objective  (of outlining clinical characteristics, classical and specific blood-based parameters and typical clinical hallmarks, in order) to identify new molecular biomarkers for the disease and the scondary objective to assess the effect of a course of DHA-TG on the ocular state and on circulating biomarkers of the above patients. Based on their results, the Authors conclude that “their study demonstrated different expression profile of cytokines/chemokines in tears from recurrent anterior NIAU patients compared with healthy controls. We confirm that inflammation is present in the quiescent stages of uveitis, being IL-6 the most outstanding cytokine in this process. We propose that IL-6 represents an important challenge as presumptive prognostic biomarker for recurrent anterior NIAU. Furthermore, this 3 month follow-up work strongly reinforces that a regular oral administration of DHA-TG reduces the inflammatory load and may potentially supply an immunomodulatory profilaxis adjunctive mediator for patients at risk of uveitis vision loss.”

Indeed, in the most benevolent hypothesis this research work can be considered an interventional pilot study whose outcomes do not allow any sensible conclusion on the inflammatory milieu of the background pathological condition, on the biomarker role of any individual cytokine  and on the antiinflammatory property of the intervention because of the lack of information concerned with the background blood level of cytokines, the lack on any attempt to sample power calculation (some 15 individuals studied per group is  very low number for any conclusion to be drawn) and for the signifcant effect of the intervention in the pathological and control group.

Fotunately, the Authors seem to be aware of the limitations of their study  and, in fact, they list a number, not complete, of limitations “Limitations of the present study derive mostly from the partial retrospective nature of this study. In fact, incomplete follow-up or insufficient registers may have led to a misestimation of data. The relatively small final sample of participants after being distributed in each subgroup reaching the 3-month follow-up, has also be considered during the statistical processing. Patients who developed exacerbation of the NIAU were excluded from data processing in order to avoid statistical power loss (3 cases). The above patients needed to be urgently treated. Relapse was not identified everytime in the follow-up visits. Systemic conditions could be underestimated despite the exhaustive anamnesis, interspecialist visits and complementary exams. Regarding the blood collection and processing, small problems have emerged (2 cases). As cytokine levels has been shown to be noticeably different between serum and plasma, but cytokine levels in plasma were reported to be more stable than in serum, the study was run in plasma from UG patients versus CG. Furthermore, as cytokines in serum and plasma are affected by multiple freeze/thaw cycles, special attention was paid to this concern by performing the cytokine assays at the same time to establish simultaneous comparison between groups and subgroups [92]." that hamper the impact of the study. “

Unfortunaltely, these same Authors are at variance with the limitations they have raised and conclude that “Luckily, none of these scarce limitations are likely to have introduced bias, so as to clearly influence the study conclusions.” This conclusion appears like a self-absolution not granted by the research (methods and results) facts.